# REAL-TIME EXPLANATIONS FOR TABULAR FOUNDATION MODELS

**Luan Borges Teodoro Reis Sena**[1,2]**, Francisco Galuppo Azevedo**[1,2]
[1]Kunumi Institute, Belo Horizonte, Minas Gerais, Brazil
[2]Universidade Federal de Minas Gerais, Belo Horizonte, Minas Gerais, Brazil
{luan.borges,francisco}@kunumi.com

## ABSTRACT

Interpretability is central for scientific machine learning, as understanding *why* models make predictions enables hypothesis generation and validation. While tabular foundation models show strong performance, existing explanation methods like SHAP are computationally expensive, limiting interactive exploration. We introduce ShapPFN, a foundation model that integrates Shapley value regression directly into its architecture, producing both predictions and explanations in a single forward pass. On standard benchmarks, ShapPFN achieves competitive performance while producing high-fidelity explanations ($R^2$=0.96, cosine=0.99) over 1000× faster than KernelSHAP (0.06s vs 610s). Our code is available at https://github.com/kunumi/ShapPFN

## 1    INTRODUCTION

In scientific applications, interpretability is essential for validating learned mechanisms and for generating testable scientific hypotheses. Tabular foundation models have shown strong performance across heterogeneous datasets (Hollmann et al., 2025; QU et al., 2025), but providing explanations remains challenging. Shapley values offer a theoretically grounded method for feature attribution, but exact computation is intractable due to the combinatorial number of feature subsets. SHAP (Lundberg & Lee, 2017) addresses this through efficient approximations, but for tabular foundational models it remains too slow for interactive scientific workflows, where researchers need real-time feedback.

ViaSHAP (Alkhatib et al., 2025) addressed this by learning predictions through Shapley value regression, using a specialized loss to enforce SHAP properties. This makes explanations part of the forward pass rather than a separate post-hoc step. However, this approach has not been applied to foundation models that generalize across diverse tabular datasets.

We introduce ShapPFN, a tabular foundation model that performs prediction via Shapley value regression. Our approach combines the generalization of Prior-Data Fitted Networks with ViaSHAP's explanation-aware prediction. Experiments show competitive accuracy on standard benchmarks with high-fidelity explanations (cosine similarity=0.99 with KernelSHAP) at over 1000× speedup (0.06s vs 610s), enabling real-time explanations.

Our main contributions are:

- We integrate Shapley value regression into tabular foundation models, producing both predictions and explanations in a single forward pass.
- We achieve competitive predictive performance with high-quality explanations (R²=0.96 vs KernelSHAP) at over 1000× speedup across diverse tabular datasets.

## 2    RELATED WORK

**Prior-Data Fitted Networks for Tabular Data**    Prior-Data Fitted Networks (PFNs) leverage pretraining on synthetic datasets to enable rapid adaptation to new tabular tasks via in-context learning.

TabPFN (Hollmann et al., 2025) uses a Transformer trained on synthetic data, achieving strong performance in small-data regimes with fast inference. TabICL (QU et al., 2025) scales this approach using structural causal models to generate more realistic feature dependencies. NanoTabPFN (Pfefferle et al., 2025) provides a lightweight reimplementation demonstrating that core PFN inductive biases can be preserved in minimal architectures. These works focus on predictive accuracy; we investigate how to add efficient feature attribution to PFN-style models.

**Explainability and Shapley-Based Methods**    Shapley values provide theoretically grounded feature attribution with desirable axiomatic properties including additivity and fairness. However, exact computation is intractable due to exponential feature coalitions. SHAP (Lundberg & Lee, 2017) introduces efficient approximations that make Shapley-consistent explanations computationally feasible, explaining its widespread adoption across scientific domains including drug discovery (Rodríguez-Pérez & Bajorath, 2020), agriculture (Monteiro et al., 2024), and climate modeling (Lu et al., 2024). ViaSHAP (Alkhatib et al., 2025) proposes jointly learning predictions and Shapley-consistent attributions within a single model, eliminating external explanation procedures. We extend this approach to foundation models that generalize across diverse tabular datasets.

Prior work has developed two complementary directions: tabular foundation models (TabPFN, TabICL) that achieve strong generalization through synthetic pretraining, and Shapley-based methods (SHAP, ViaSHAP) that provide principled feature attributions. However, no existing approach combines both. We bridge this gap by integrating Shapley value regression into the foundation model architecture, achieving both computational efficiency and predictive performance while enabling real-time explanations.

## 3 SHAPPFN

### 3.1 ARCHITECTURE

ShapPFN follows the same TabPFN-style design of alternating attention over features and datapoints, but instead of decoding only from the test target token embedding, it decomposes the prediction into a base term plus per-feature additive contributions. The architecture is based on nanoTabPFN (Pfefferle et al., 2025), with the core transformer blocks unchanged. The key modifications are the addition of two specialized decoder heads: BaseDecoder for the global baseline and ShapDecoder for per-feature contributions.

The **FeatureEncoder** creates embeddings for all values in X by normalizing each feature (using mean and standard deviation), clipping extreme values, and mapping to a high-dimensional vector via a linear layer. The embeddings for the target variable are created by the **TargetEncoder**, which applies a linear layer to create target embeddings. The embeddings are then passed to a stack of **TransformerEncoderLayers** sequentially, which apply attention between features, followed by attention between datapoints, in this step, the training data can attend to itself but not to the test data.

After that, a **BaseDecoder** produces a global baseline prediction conditioned on the mean embedding of trained target tokens, yielding a shared base logit vector for all the test rows. A **ShapDecoder** is then applied to the test feature embeddings to produce per-feature per-class contributions $\phi$. The final logits are then computed as the sum of the feature contributions plus the base term:

$$f_\theta(x) = base + \sum_{f=1}^{F} \phi_f(x)$$

This design makes the prediction explicitly additive over features, enabling a SHAP-like decomposition directly from the architecture.

### 3.2 LOSS

The loss is based on ViaSHAP: masked features should have Shapley value 0, while predictions based on a subset of features should equal the sum of their corresponding Shapley contributions.

Figure 1: ShapPFN architecture. The model encodes features and targets into embeddings, processes them through transformer layers with alternating attention over features and datapoints, then decodes into a global baseline and per-feature contributions. The final prediction is explicitly additive over features, enabling SHAP-like decomposition directly from the architecture.

In our formulation, we first perform an unmasked forward pass to obtain the model's base term and per-feature Shapley contributions. We then sample $S$ feature coalitions with sizes $|s| \in \{1, \ldots, F-1\}$, where $F$ is the number of features. For each coalition $s$, we construct a masked input by replacing all excluded features with values drawn from $K$ randomly selected rows of the dataset. This masking procedure corresponds to the interventional Shapley definition (Janzing et al., 2020), where expectations are taken over the marginal distribution of the masked features. For each sampled coalition $s$, we obtain two estimates of the model output: a Monte Carlo approximation from $K$ forward passes, and an additive approximation from the learned Shapley values:

$$\hat{f}_\theta(x^s) = \frac{1}{K} \sum_{k=1}^{K} f_\theta(x_k^s) \qquad \bar{f}_\theta(x^s) = \text{base}(x) + \sum_{j \in s} \phi_j(x).$$

The Shapley consistency loss is the Shapley-kernel-weighted mean squared error between these estimates, during training, we add this Shapley consistency loss to the standard cross-entropy loss, with a weighting coefficient controlling its relative contribution.

$$\mathcal{L}_{\text{shap}} = \frac{1}{S} \sum_{s=1}^{S} w_s \left( \bar{f}_\theta(x^s) - \hat{f}_\theta(x^s) \right)^2, \qquad w_s = \frac{(F-1)}{\binom{F}{|s|} |s| \, (F - |s|)}.$$

## 4 EXPERIMENTS

### 4.1 EXPERIMENTAL SETUP

ShapPFN was trained on 256,000 TabICL-generated synthetic datasets for 8,000 steps (batch size 32, schedule-free AdamW (Loshchilov & Hutter, 2019)). Hyperparameters were tuned via ROC-AUC on eight validation datasets. Training ShapPFN incurs higher computational cost than NanoTabPFN due to the additional SHAP loss, but this overhead affects only pretraining and not inference. More details are added in Appendix A.

### 4.2 PERFORMANCE ACROSS TASKS

**Datasets.** We evaluate on the OpenML-CC18 benchmark suite (Bischl et al., 2021), which comprises a diverse collection of classification tasks commonly used for benchmarking automated machine learning and classical learning algorithms.

**Baselines.** We compare ShapPFN to state-of-the-art and classical ML models, plus nanoTabPFN trained identically but without the SHAP loss, with the same lr optimization scheme. This provides a fair comparison as both models have the same base architecture and training data. For multi-class tasks, both use a OneVsAll (OVA) approach.

Table 1 shows that ShapPFN(+loss) achieves an average ROC-AUC of 0.848 across all datasets, matching NanoTabPFN (0.848) despite additional SHAP constraints. ShapPFN(arch) without SHAP loss scores 0.837, indicating a small architectural cost. While TabPFN v2 leads at 0.872, ShapPFN remains competitive with Random Forest (0.851) and outperforms other classical baselines.

| Dataset | Dataset size | | Foundation models | | | | Classical models | | |
|---|---|---|---|---|---|---|---|---|---|
| | $n$ | $d$ | TabPFN v2 | Nano TabPFN | ShapPFN (arch) | ShapPFN (+loss) | Random Forest | KNN | Decision Tree |
| *HPO datasets* | | | | | | | | | |
| banknote | 1,372 | 5 | **1.000** | 0.999 | 0.993 | 0.991 | 0.995 | **1.000** | 0.938 |
| blood-transf | 748 | 5 | 0.694 | 0.696 | 0.698 | **0.703** | 0.674 | 0.625 | 0.624 |
| diabetes | 768 | 9 | 0.844 | **0.852** | 0.851 | 0.835 | 0.819 | 0.751 | 0.656 |
| electricity | 45,312 | 9 | **0.866** | 0.860 | 0.856 | 0.863 | 0.843 | 0.650 | 0.680 |
| ilpd | 583 | 11 | 0.737 | **0.740** | 0.728 | 0.724 | 0.727 | 0.651 | 0.558 |
| phoneme | 5,404 | 6 | 0.849 | 0.822 | 0.820 | 0.827 | **0.850** | 0.796 | 0.643 |
| tic-tac-toe | 958 | 10 | 0.827 | 0.689 | 0.702 | 0.722 | **0.899** | 0.855 | 0.729 |
| wilt | 4,839 | 6 | **0.987** | 0.927 | 0.842 | 0.852 | 0.925 | 0.782 | 0.714 |
| **Average (HPO)** | | | **0.851** | **0.823** | **0.811** | **0.815** | **0.842** | **0.764** | **0.693** |
| *Eval-only datasets* | | | | | | | | | |
| analcat-auth | 841 | 71 | **1.000** | 0.999 | 0.998 | 0.999 | 0.998 | 0.988 | 0.865 |
| analcat-dmft | 797 | 5 | **0.598** | 0.581 | 0.586 | 0.580 | 0.538 | 0.571 | 0.506 |
| balance | 625 | 5 | **0.991** | 0.966 | 0.938 | 0.955 | 0.821 | 0.858 | 0.708 |
| bank-marketing | 45,211 | 17 | **0.777** | 0.735 | 0.701 | 0.744 | 0.742 | 0.628 | 0.618 |
| car | 1,728 | 7 | **0.980** | 0.935 | 0.931 | 0.927 | 0.949 | 0.802 | 0.740 |
| churn | 5,000 | 21 | 0.869 | 0.879 | **0.881** | 0.865 | 0.864 | 0.510 | 0.721 |
| climate-crash | 540 | 21 | **0.974** | 0.923 | 0.891 | 0.932 | 0.892 | 0.819 | 0.657 |
| cmc | 1,473 | 10 | **0.744** | 0.708 | 0.714 | 0.715 | 0.696 | 0.614 | 0.581 |
| connect-4 | 67,557 | 43 | 0.576 | 0.566 | 0.538 | 0.569 | **0.578** | 0.473 | 0.546 |
| credit-g | 1,000 | 21 | 0.810 | 0.762 | 0.743 | 0.822 | **0.851** | 0.527 | 0.738 |
| first-order | 6,118 | 52 | 0.666 | **0.697** | 0.675 | 0.683 | 0.668 | 0.640 | 0.541 |
| gesture-phase | 9,873 | 33 | **0.753** | 0.677 | 0.709 | 0.720 | 0.732 | 0.727 | 0.582 |
| jungle-chess | 44,819 | 7 | 0.860 | 0.834 | 0.832 | 0.836 | **0.869** | 0.760 | 0.626 |
| kc1 | 2,109 | 22 | 0.762 | 0.769 | **0.774** | 0.752 | 0.730 | 0.684 | 0.537 |
| kc2 | 522 | 22 | 0.762 | 0.763 | **0.770** | 0.752 | 0.712 | 0.638 | 0.534 |
| kr-vs-kp | 3,196 | 37 | **0.992** | 0.941 | 0.906 | 0.962 | 0.985 | 0.850 | 0.919 |
| mfeat-four | 2,000 | 77 | **0.983** | 0.946 | 0.935 | 0.945 | 0.963 | 0.945 | 0.803 |
| mfeat-kar | 2,000 | 65 | **0.997** | 0.989 | 0.985 | 0.991 | 0.990 | 0.985 | 0.828 |
| mfeat-morph | 2,000 | 7 | **0.962** | 0.950 | 0.946 | 0.954 | 0.942 | 0.854 | 0.825 |
| mfeat-zern | 2,000 | 48 | **0.978** | 0.967 | 0.963 | 0.959 | 0.972 | 0.951 | 0.761 |
| numerai28.6 | 96,320 | 22 | 0.538 | 0.568 | **0.576** | 0.566 | 0.559 | 0.532 | 0.500 |
| optdigits | 5,620 | 65 | **0.992** | 0.978 | 0.977 | 0.991 | 0.992 | 0.980 | 0.822 |
| ozone-level-8hr | 2,534 | 73 | **0.912** | 0.866 | 0.806 | 0.860 | 0.872 | 0.543 | 0.542 |
| pc1 | 1,109 | 22 | 0.672 | 0.657 | **0.674** | 0.632 | 0.630 | 0.392 | 0.627 |
| pc3 | 1,563 | 38 | 0.743 | 0.746 | **0.781** | 0.763 | 0.728 | 0.628 | 0.519 |
| pc4 | 1,458 | 38 | **0.890** | 0.813 | 0.762 | 0.819 | 0.819 | 0.533 | 0.692 |
| pendigits | 10,992 | 17 | **0.997** | 0.985 | 0.973 | 0.974 | 0.991 | 0.985 | 0.857 |
| phishing | 11,055 | 31 | 0.974 | 0.967 | 0.958 | 0.969 | **0.976** | 0.907 | 0.877 |
| qsar-biodeg | 1,055 | 42 | **0.898** | 0.864 | 0.864 | 0.864 | 0.868 | 0.770 | 0.731 |
| satimage | 6,430 | 37 | **0.978** | 0.969 | 0.965 | 0.962 | 0.971 | 0.940 | 0.826 |
| segment | 2,310 | 20 | **0.985** | 0.972 | 0.963 | 0.973 | 0.979 | 0.947 | 0.906 |
| spambase | 4,601 | 58 | **0.968** | 0.946 | 0.914 | 0.952 | 0.952 | 0.690 | 0.847 |
| splice | 3,190 | 62 | **0.988** | 0.965 | 0.898 | 0.981 | 0.978 | 0.889 | 0.863 |
| steel-plates | 1,941 | 28 | **0.916** | 0.879 | 0.881 | 0.865 | 0.896 | 0.670 | 0.742 |
| vehicle | 846 | 19 | **0.943** | 0.911 | 0.887 | 0.879 | 0.903 | 0.795 | 0.736 |
| wall-robot | 5,456 | 25 | **0.994** | 0.927 | 0.885 | 0.945 | 0.994 | 0.790 | 0.920 |
| wdbc | 569 | 31 | **0.997** | 0.990 | 0.986 | 0.988 | 0.983 | 0.961 | 0.953 |
| **Average (Eval-only)** | | | **0.876** | **0.854** | **0.842** | **0.855** | **0.854** | **0.751** | **0.719** |
| **Average (All)** | | | **0.872** | **0.848** | **0.837** | **0.848** | **0.851** | **0.753** | **0.714** |

Table 1: **ROC-AUC benchmark across tabular datasets.** Foundation models are separated from classical baselines. Datasets are split into *HPO* and *Eval-only* subsets, with per-group averages. ShapPFN(+loss) is the proposed full method; ShapPFN(arch) is the same architecture with loss weight set to 0. Best score per row is in bold and highlighted in gold; second/third best are highlighted in silver/bronze.

| Dataset | Time | | | ShapPFN(+loss) | | | ShapPFN(arch) | | |
|---|---|---|---|---|---|---|---|---|---|
| | Kernel (s) | Model (s) | Speedup | $R^2$ | Cosine | Spearman | $R^2$ | Cosine | Spearman |
| *HPO datasets* | | | | | | | | | |
| banknote | 3.6 | 0.03 | 105× | 0.965 | 0.989 | 0.924 | 0.596 | 0.936 | 0.819 |
| blood-transf | 3.5 | 0.03 | 104× | 0.939 | 0.988 | 0.970 | -0.826 | 0.698 | 0.607 |
| diabetes | 126 | 0.04 | 3179× | 0.984 | 0.994 | 0.959 | 0.324 | 0.882 | 0.745 |
| electricity | 130 | 0.04 | 3080× | 0.975 | 0.990 | 0.975 | 0.679 | 0.906 | 0.794 |
| phoneme | 6.9 | 0.04 | 182× | 0.965 | 0.986 | 0.968 | 0.228 | 0.831 | 0.785 |
| wilt | 7.2 | 0.03 | 219× | 0.952 | 0.985 | 0.965 | -0.024 | 0.534 | 0.447 |
| **Average (HPO)** | **46** | **0.04** | **402×** | **0.963** | **0.989** | **0.960** | **0.163** | **0.798** | **0.699** |
| *Eval-only datasets* | | | | | | | | | |
| analcat-dmft | 187 | 0.12 | 1574× | 0.971 | 0.990 | 0.982 | -0.462 | 0.632 | 0.833 |
| balance | 6.5 | 0.07 | 87.6× | 0.980 | 0.992 | 0.987 | 0.367 | 0.873 | 0.859 |
| car | 4016 | 0.08 | 50063× | 0.956 | 0.981 | 0.893 | 0.210 | 0.669 | 0.675 |
| cmc | 380 | 0.06 | 6639× | 0.970 | 0.989 | 0.965 | 0.436 | 0.810 | 0.695 |
| ilpd | 1132 | 0.04 | 29154× | 0.919 | 0.970 | 0.960 | 0.126 | 0.604 | 0.474 |
| jungle-chess | 31 | 0.06 | 514× | 0.980 | 0.993 | 0.986 | 0.459 | 0.824 | 0.773 |
| mfeat-morph | 102 | 0.19 | 542× | 0.964 | 0.985 | 0.980 | 0.119 | 0.713 | 0.838 |
| tic-tac-toe | 2414 | 0.05 | 48888× | 0.958 | 0.983 | 0.848 | 0.270 | 0.588 | 0.271 |
| **Average (Eval-only)** | **1034** | **0.08** | **3408×** | **0.962** | **0.985** | **0.950** | **0.191** | **0.714** | **0.677** |
| **Average (All)** | **610** | **0.06** | **1364×** | **0.963** | **0.987** | **0.954** | **0.179** | **0.750** | **0.687** |

Table 2: **SHAP values quality and computational efficiency.** SHAP approximation quality and computation time across datasets. Quality metrics: $R^2$, Cosine similarity, Spearman correlation (vs KernelSHAP). ShapPFN(+loss) is the full model with SHAP loss; ShapPFN(arch) uses the same architecture without SHAP loss. SHAP values are computed using a background set of 150 samples and explanations are generated for 50 test instances per dataset. Times in seconds; speedup uses geometric mean.

## 4.3 EXPLAINABILITY EVALUATION

Table 2 compares SHAP quality and computational efficiency. We adopt KernelSHAP as the explainability baseline because it is a well-established, model-agnostic method for approximating Shapley values and is widely used as a reference standard when assessing attribution faithfulness. For this evaluation, we restrict KernelSHAP comparisons to datasets with a low number of features, as its computational cost becomes prohibitive for higher-dimensional settings. ShapPFN(+loss) achieves high agreement with KernelSHAP ($R^2$=0.963, cosine similarity=0.987, Spearman=0.954) while delivering over three orders of magnitude speedup (610s → 0.06s average, up to 50000× on some datasets). This performance generalizes across both HPO and Eval-only datasets. Critically, ShapPFN(arch) without SHAP loss shows substantially degraded explanation quality (R²=0.179, cosine=0.750), demonstrating that the SHAP loss is essential for faithful explanations. These results confirm that ShapPFN produces KernelSHAP-quality explanations in real-time.

## 5 CONCLUSION

We introduced ShapPFN, integrating Shapley value regression into tabular foundation models to enable real-time explanations. ShapPFN matches baseline predictive performance (0.848 ROC-AUC) while producing high-fidelity explanations ($R^2$=0.963 vs KernelSHAP) at over 1000× speedup (0.06s vs 610s). This advancement transforms SHAP from a post-hoc tool into an interactive component suitable for scientific workflows requiring immediate feedback. The ability to obtain principled feature attributions without compromising accuracy or efficiency opens new possibilities for interpretable foundation models in domains where transparent model behavior is essential for trust and discovery.

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

## A HYPERPARAMETER INFLUENCE

We trained ShapPFN on 256,000 synthetic datasets generated by TabICL's data generation pipeline. Training ran for 8,000 optimization steps with batch size 32. All datasets were binary classification tasks with 2–5 input features and maximum sequence length 200 samples. For each dataset, we randomly sampled the training split size between 10% and 90% of available samples. We used the schedule-free AdamW optimizer without weight decay (Loshchilov & Hutter, 2019). We tuned

| Parameter Optimized | Loss Wt. | Warmup Steps | Num Subsets | Bkg Samples | SHAP Cos | ROC AUC |
|---|---|---|---|---|---|---|
| SHAP Loss Weight | 1 | 2000 | 4 | 4 | 0.979 | **0.829** |
| | 2 | 2000 | 4 | 4 | 0.977 | 0.826 |
| | 10 | 2000 | 4 | 4 | 0.979 | 0.798 |
| | 50 | 2000 | 4 | 4 | 0.890 | 0.777 |
| Warmup Steps | 1 | 0 | 4 | 4 | 0.977 | 0.830 |
| | 1 | 400 | 4 | 4 | 0.979 | 0.828 |
| | 1 | 1200 | 4 | 4 | 0.988 | **0.835** |
| | 1 | 2000 | 4 | 4 | 0.979 | 0.829 |
| Num Subsets | 1 | 1200 | 2 | 4 | 0.987 | 0.834 |
| | 1 | 1200 | 4 | 4 | 0.988 | **0.835** |
| | 1 | 1200 | 8 | 4 | 0.987 | 0.835 |
| Bkg Samples | 1 | 1200 | 4 | 2 | 0.981 | 0.833 |
| | 1 | 1200 | 4 | 4 | 0.988 | 0.835 |
| | 1 | 1200 | 4 | 8 | 0.989 | **0.836** |

Table 3: **Sequential hyperparameter optimization.** Each block optimizes one parameter (highlighted in gray) while keeping others fixed. Best ROC AUC in each block shown in bold.

hyperparameters (learning rate and ViaSHAP-specific parameters) based on ROC-AUC performance on eight validation datasets.

For the learning rate optimization, we evaluated the values $\{5 \times 10^{-4}, 10^{-3}, 2 \times 10^{-3}, 4 \times 10^{-3}, 8 \times 10^{-3}, 1.6 \times 10^{-2}, 3.2 \times 10^{-2}\}$, with both ShapPFN and NanoTabPFN achieving their best performance at $2 \times 10^{-3}$. Both models have 3 layers, 4 attention heads, an embedding size of 96, and a hidden layer size of 192.

Table 3 reports the sequential hyperparameter optimization for the SHAP specific parameters. We selected the model with the highest ROC-AUC that maintained high cosine similarity ($> 0.95$) to KernelSHAP values. Increasing the SHAP loss weight beyond 1 yields diminishing returns and degrades both cosine similarity and ROC-AUC. Warmup steps have a moderate effect, with 1200 steps giving the best performance. The number of SHAP subsets has minor impact, with performance saturating at four subsets. Increasing background samples yields small improvements, with the best result at eight samples. These results align with ViaSHAP findings, where the number of samples affects performance less than the loss weight.

In our setup, pretraining ShapPFN required approximately 3000 seconds, compared to 250 seconds for NanoTabPFN under identical settings, this overhead is incurred only once during offline training, inference time remains unchanged.

