# OpenReview forum: "[Short] Real-Time Explanations for Tabular Foundation Models"
_ICLR.cc/2026/Workshop/FM4Science — ICLR 2026 Workshop FM4Science Poster_

### Official Review · Reviewer_Bb6d · 2026-02-19
**Promising real-time SHAP-style explanations for PFNs, but evaluation and pretraining assumptions need clarification**

**Rating:** 6
**Confidence:** 3

**Review:**

This paper proposes ShapPFN, a tabular foundation model that integrates SHAP-style additive explanations directly into a PFN architecture. The idea is intuitive: the model predicts a baseline plus per-feature contributions in a single forward pass, trained with a ViaSHAP-style consistency loss. The motivation — enabling real-time explanations for scientific workflows — is compelling, and the paper is generally clear and easy to follow.

I think the main strength of the work is its practical angle. Integrating explanation outputs into the model itself is a neat engineering idea, and the reported speedups over KernelSHAP are impressive. The architecture is also relatively simple, which makes the approach easy to understand.

My main concerns are about how explanation quality is evaluated and about the pretraining setup.

First, most of the explanation evaluation is based on agreement with KernelSHAP. While this is a reasonable baseline, it feels somewhat circular because the training objective already encourages SHAP-like behavior under a similar masking procedure. High agreement with KernelSHAP does not necessarily mean the explanations are faithful or scientifically meaningful. I would have liked to see at least one additional type of evaluation — for example feature removal tests, synthetic ground-truth attribution checks, or some analysis of whether the explanations align with known structure in data.

Second, Appendix A mentions that the synthetic pretraining tasks use only 2–5 features, while many evaluation datasets contain several dozens of features. It is not clear how a model pretrained on such low-dimensional tasks learns explanation behavior that transfers to higher-dimensional settings. Since explanation quality is a central claim of the paper, this mismatch deserves more discussion or analysis.

Finally, the authors mention that code will be released upon acceptance, which I think will be important for understanding implementation details and assessing reproducibility.

Overall, I think this is a nice workshop contribution with a clear idea and promising efficiency gains, but the explanation evaluation and the pretraining assumptions could be strengthened to make the claims more convincing.

---

### Official Review · Reviewer_DQbD · 2026-02-23
**Novel and solid work**

**Rating:** 7
**Confidence:** 4

**Review:**

This manuscript introduces a novel tabular foundation model (ShapPFN) that integrates Shapley value regression into a PFN-style architecture. On OpenML-CC18, ShapPFN maintains high ROC-AUC, produces highly interpretable outputs, and significantly improves efficiency.

#### **Strengths**
1. It's quite interesting to combine two approaches (a foundational tabular model with exceptional generalisation capabilities, and a Shapley-based interpretative method offering reliable feature attribution) demonstrating both novelty and logical coherence.
2. Experiments are complete and solid.
3. ShapPFN achieves over 1000-fold acceleration in interpretation generation, enabling real-time scientific exploration.

#### **Weaknesses**
No empirical comparison to amortized SHAP methods such as FastSHAP or InstaSHAP, or to recent functional-decomposition explainers (*e.g.*, STRIDE) that can also operate efficiently.

---

### Meta-Review · Area_Chair_zrzF · 2026-02-27

**Recommendation:** Accept (Poster)
**Confidence:** 4

**Metareview:**

This submission has received two reviews. One reviewer rated this submission as "accept," and another reviewer rated this work as "Marginally above acceptance threshold."

After having read the reviews, I recommend this paper for "acceptance" and ask the authors to implement the feedback of all reviews to the camera ready version of the paper.

---

### Decision · Program_Chairs · 2026-03-03

Accept (Poster)